# Probe-Seq enables transcriptional profiling of specific cell types from heterogeneous tissue by RNA-based isolation

Ryoji Amamoto[1,2], Mauricio D Garcia[1,2], Emma R West[1,2], Jiho Choi[1,2], Sylvain W Lapan[1,2], Elizabeth A Lane[1], Norbert Perrimon[1], Constance L Cepko[1,2]*

[1]Department of Genetics, Blavatnik Institute, Howard Hughes Medical Institute, Harvard Medical School, Boston, United States; [2]Department of Ophthalmology, Blavatnik Institute, Howard Hughes Medical Institute, Harvard Medical School, Boston, United States

**Abstract** Recent transcriptional profiling technologies are uncovering previously-undefined cell populations and molecular markers at an unprecedented pace. While single cell RNA (scRNA) sequencing is an attractive approach for unbiased transcriptional profiling of all cell types, a complementary method to isolate and sequence specific cell populations from heterogeneous tissue remains challenging. Here, we developed Probe-Seq, which allows deep transcriptional profiling of specific cell types isolated using RNA as the defining feature. Dissociated cells are labeled using fluorescent in situ hybridization (FISH) for RNA, and then isolated by fluorescent activated cell sorting (FACS). We used Probe-Seq to purify and profile specific cell types from mouse, human, and chick retinas, as well as from *Drosophila* midguts. Probe-Seq is compatible with frozen nuclei, making cell types within archival tissue immediately accessible. As it can be multiplexed, combinations of markers can be used to create specificity. Multiplexing also allows for the isolation of multiple cell types from one cell preparation. Probe-Seq should enable RNA profiling of specific cell types from any organism.

*For correspondence:
cepko@genetics.med.harvard.edu

**Competing interests:** The authors declare that no competing interests exist.

## Introduction

Multicellular eukaryotic tissues often comprise many different cell types, commonly classified using their morphological features, physiological functions, anatomical locations, and/or molecular markers. For example, the retina, a thin sheet of neural tissue in the eye that processes and transmits visual information to the brain, contains seven major cell classes – rods, cones, bipolar cells (BC), amacrine cells (AC), horizontal cells (HC), Müller glia (MG), and retinal ganglion cells (RGC), first defined primarily using morphology (*Sanes and Zipursky, 2010*; *Vlasits et al., 2019*). More recently, scRNA profiling technologies have led to the appreciation of many subtypes of these major cell classes, bringing the total number of retinal cell types close to 100 (*Macosko et al., 2015*; *Rheaume et al., 2018*; *Shekhar et al., 2016*). Such accelerated discovery of cellular diversity is not unique to the retina, as scRNA profiling is being carried out in many tissues and organisms (*Tabula Muris Consortium et al., 2018*; *Han et al., 2018*; *Regev et al., 2017*).

Several approaches have been used to transcriptionally profile tissues. Bulk RNA sequencing of whole tissues can be done at great depth, but does not capture the diversity of individual transcriptomes and often fails to reflect signatures of rare cell types. Currently, bulk sequencing of specific cell types is limited by the availability of cell type-specific promoters, enhancers, dyes, or antigens for their isolation (*Arlotta et al., 2005*; *Matsuda and Cepko, 2007*; *Molyneaux et al., 2015*;

*Siegert et al., 2012*; *Telley et al., 2016*; *Xu et al., 2018*). This has limited bulk RNA sequencing primarily to select cell types in genetically-tractable organisms. Single cell and single nucleus RNA sequencing methods have allowed for the recording of transcriptional states of many individual cells simultaneously (*Macosko et al., 2015*; *Shekhar et al., 2016*; *Jaitin et al., 2014*; *Klein et al., 2015*; *Picelli et al., 2013*; *Shalek et al., 2013*). Despite the undeniable appeal of scRNA sequencing, capturing deep profiles of specific cell populations in bulk can be sufficient or preferable for many experiments, for example when the goal is to understand the results of perturbations.

We and others have used antibodies to enable FACS-based isolation for transcriptional profiling of specific cell populations (*Molyneaux et al., 2015*; *Amamoto et al., 2019*; *Hrvatin et al., 2014*; *Pan et al., 2011*; *Pechhold et al., 2009*; *Yamada et al., 2010a*). However, antibodies are frequently unavailable for a specific cell type. Furthermore, marker proteins in certain cell types, such as neurons, are often localized to processes that are lost during tissue dissociation. We therefore aimed to create a method that would leverage the newly discovered RNA expression patterns for the isolation of specific cell populations from any organism. This led us to develop Probe-Seq, which uses a FISH method based upon a new probe design, Serial Amplification By Exchange Reaction (SABER) (*Kishi et al., 2019*). Probe-Seq uses RNA markers expressed in specific cell types to label cells for isolation by FACS and subsequent transcriptional profiling. Although specific cells cultured in vitro have been successfully labeled by FISH for isolation using FACS, this method had not yet been tested for use in tissue (*Klemm et al., 2014*; *Maeda et al., 2016*; *Yamada et al., 2010b*). We used Probe-Seq to isolate rare bipolar cells from the mouse retina, cell types that were previously defined using scRNA sequencing (*Shekhar et al., 2016*). We demonstrate that probe sets for multiple genes can be hybridized at once, allowing isolation of multiple cell types simultaneously. Moreover, the fluorescent oligos used to detect the probe sets can be quickly hybridized and then stripped. This enables isolation of an indefinite number of cell types from one sample by serial sorting and re-labeling. We extended Probe-Seq to specific bipolar cell subtypes in frozen archival human retina by labeling nuclear RNA. To further test the utility of Probe-Seq in non-vertebrate animals and non-CNS tissues, we profiled intestinal stem cells from the *Drosophila* gut. In each of these experiments, the transcriptional profiles of isolated populations closely matched those obtained by scRNA sequencing, and in most cases, the number of genes detected exceeded 10,000. Finally, we used Probe-Seq on the chick retina, an organism that is difficult to genetically manipulate, to determine the transcriptional profile of a subset of developing retinal cells that give rise to the chick high acuity area. Taken together, Probe-Seq is a method that enables deep transcriptional profiling of specific cell types in heterogeneous tissue from potentially any organism.

## Results

### Specific bipolar cell subtypes can be isolated and profiled from the mouse retina using Probe-Seq

To determine whether Probe-Seq can enable the isolation and profiling of specific cell types based upon FISH labeling, we tested it using the mouse retina. The retina is a highly heterogeneous tissue, with cell classes and subtypes classified by scRNA sequencing, as well as more classical methods (*Vlasits et al., 2019*). We used a new method for FISH, SABER-FISH, to label the intracellular RNA (*Kishi et al., 2019*). SABER-FISH uses OligoMiner to design 20–40 nt oligonucleotides (oligos) that are complementary to the RNA species of interest and are optimized for minimal off-target binding (*Beliveau et al., 2018*). The oligos are pooled and extended using a Primer Exchange Reaction (*Kishi et al., 2018*), which appends many copies of a short-repeated sequence (concatemers) to each oligo in the set. This pooled, extended oligo preparation will be referred to as a gene-specific probe set. To allow for detection of multiple gene-specific probe sets, the concatemer sequences can be made unique for each probe set. The concatemers can then be detected by the hybridization of fluorescent oligos.

To isolate specific BC subtypes, fresh adult mouse retinas were dissociated, fixed, and permeabilized prior to FISH labeling (*Figure 1a*). We designed gene-specific probe sets against *Vsx2*, a marker of all BCs and MG, and *Grik1*, a marker of BC2, BC3A, BC3B, and BC4 subtypes (~2% of all retinal cells) (*Shekhar et al., 2016*) (*Figure 1b*). *Vsx2* and *Grik1* probe sets were hybridized to the dissociated retinal cells overnight at 43°C, and fluorescent oligos were subsequently hybridized to

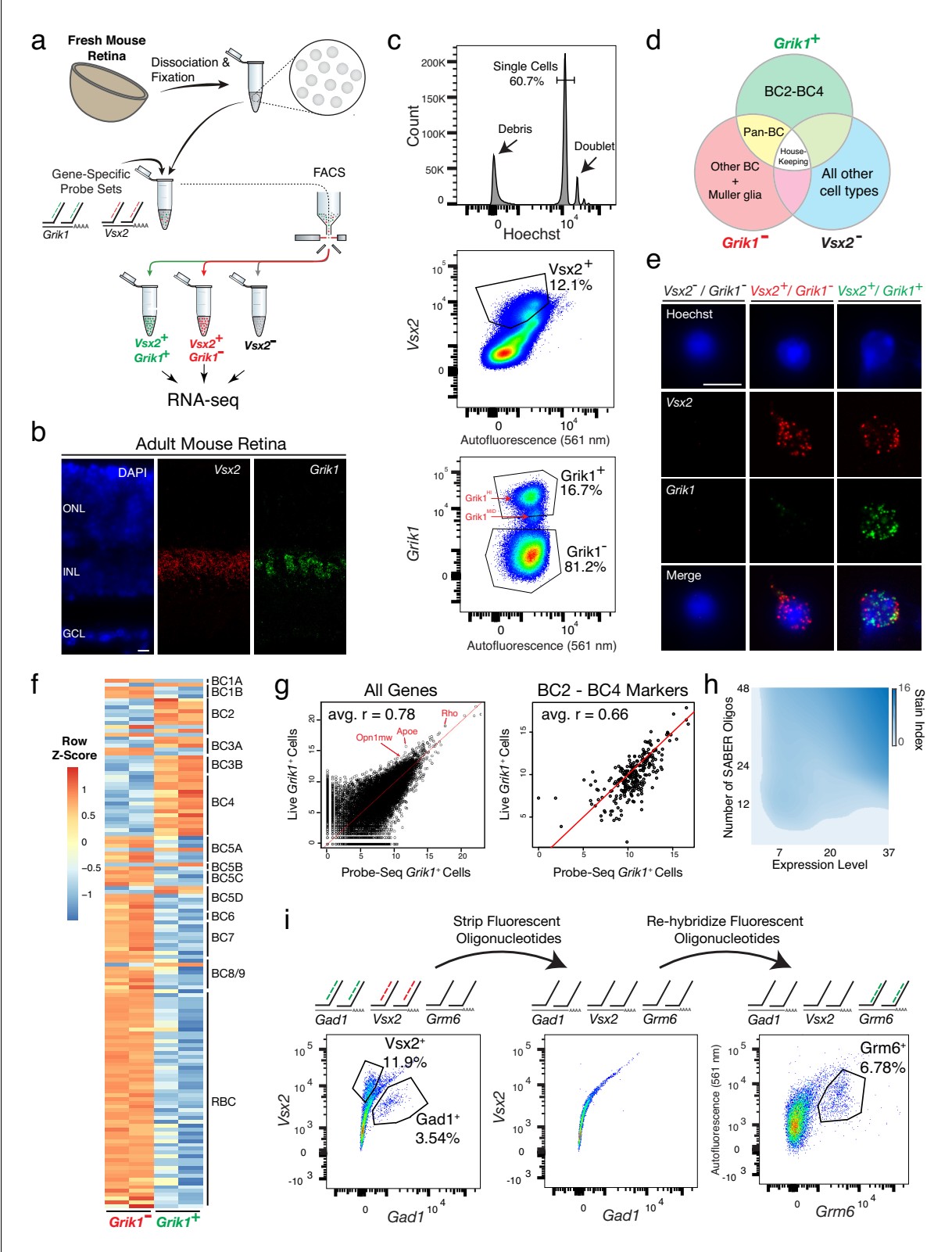

**Figure 1.** Isolation and transcriptional profiling of specific BC subtypes from the adult mouse retina. (**a**) Schematic of Probe-Seq for the adult mouse retina. The retina was dissociated into single cells, fixed, and permeabilized. Cells were incubated with gene-specific probe sets against *Vsx2* and *Grik1* and subsequently incubated with fluorescent oligos. Three populations of labeled cells (*Vsx2⁻*, *Vsx2⁺/Grik1⁻*, and *Vsx2⁺/Grik1⁺*) were isolated by FACS for downstream RNA sequencing. (**b**) SABER-FISH signals from an adult mouse retina section using *Vsx2* and *Grik1* probe sets. (**c**) Representative FACS

*Figure 1 continued on next page*

*Figure 1 continued*

plot of all events (top panel) on a Hoechst histogram. The debris is the peak near 0. The first peak after the debris is the single cell 2N peak. 4N doublets and other cell clumps are in the peaks to the right of the single cell peak. Representative FACS plots of all single cells (middle panel) with *Vsx2* fluorescence on the y-axis and autofluorescence (561 nm) on the x-axis. The negative population ran along the diagonal. The *Vsx2*+ population (12.1%) was left shifted, indicating high *Vsx2* fluorescence and low autofluorescence. FACS plot of only the *Vsx2*+ population (bottom panel) with *Grik1* fluorescence on the y-axis and autofluorescence (561 nm) on the x-axis. *Vsx2*+/*Grik1*+ population (16.7%) displayed strong separation from the *Vsx2*+/ *Grik1*- population (81.2%). The *Grik1*^MID population was included in the *Vsx2*+/*Grik1*+ population. (**d**) Expected distribution of retinal cell type markers expressed in each isolated population. (**e**) Images of dissociated mouse retinal cells after the SABER-FISH protocol on dissociated mouse retinal cells before FACS. (**f**) A heatmap representing relative expression levels of BC subtype markers previously identified by scRNA sequencing that are differentially expressed (adjusted p-value<0.05) between *Grik1*- and *Grik1*+ populations. (**g**) A representative scatter plot of log$_2$ normalized counts of all genes (left panel) or BC2 – BC4 marker genes (right panel) between Live *Grik1*+ cells and Probe-Seq *Grik1*+ Cells. Red line indicates a slope of 1. Select cell class-specific markers (Opn1mw (cones), Apoe (MG), and Rho (rods)) are labeled in red. (**h**) A heatmap of the stain index with varying levels of transcript expression and number of oligos. The boundary indicates a cutoff of SI < 2. (**i**) Schematic and flow cytometry plots of iterative Probe-Seq. Three gene-specific probe sets (*Gad1, Vsx2,* and *Grm6*) were hybridized to dissociated mouse retinal cells, and fluorescent oligos were hybridized only to *Gad1* and *Vsx2* probe sets to detect subsets of ACs (*Gad1*+; 3.54%) and BC/MG (*Vsx2*+; 11.9%). The fluorescent oligos were subsequently stripped with 50% formamide, which abolished the staining based on flow cytometry (middle panel). Fluorescent oligos for *Grm6* were then hybridized to label a subset of BCs (right panel; *Grm6*+; 6.78%). HC, Horizontal Cell; RGC, Retinal Ganglion Cell; AC, Amacrine Cell; BC, Bipolar Cell; MG, Müller Glia; ONL, Outer Nuclear Layer; INL, Inner Nuclear Layer; GCL, Ganglion Cell Layer. Scale bars: 10 μm (**b, e**).

The online version of this article includes the following figure supplement(s) for figure 1:

**Figure supplement 1.** Quality control of mouse Probe-Seq RNA sequencing.
**Figure supplement 2.** BC markers are enriched in the *Vsx2*+/*Grik1*+ and *Vsx2*+/*Grik1*- populations.
**Figure supplement 3.** Validation of a *Grik1*+ population-enriched transcript, *Tpbgl,* by SABER-FISH.
**Figure supplement 4.** Gene body coverage of Live and Probe-Seq cells show 3′ bias in the Probe-Seq population.
**Figure supplement 5.** Probe-Seq stain index is correlated with the number of oligos and the expression level.

the gene-specific probe sets. By FACS, single cells were identified by gating for a single peak of Hoechst+ events, while debris and doublets were excluded (*Figure 1c*). Out of these single cells, the *Vsx2*+ population was judged to be the cells that shifted away from the diagonal *Vsx2*- events (*Figure 1c*). Out of the *Vsx2*+ population, we found three populations that were *Grik1*-, *Grik1*^MID, and *Grik1*^HI. Based upon scRNA sequencing of BC subtypes, *Grik1*^MID likely corresponded to BC2, and *Grik1*^HI to BC3A, BC3B, and BC4 (*Shekhar et al., 2016*). We isolated both *Grik1*^MID and *Grik1*^HI (henceforth called *Grik1*+) cells as well as *Vsx2*+/*Grik1*- (henceforth called *Grik1*-) and *Vsx2*- cell populations. We expected the *Grik1*+ population to contain BC2 – BC4, the *Grik1*- population to contain other BC subtypes and MG, and the *Vsx2*- population to contain non-BC/MG cell types (*Figure 1d*). The isolated populations displayed the expected FISH puncta (*Figure 1e*). To determine the purity of populations isolated using FACS, based on their SABER signal, the cells were placed on a microscope slide after FACS, and the percentage of cells that had *Vsx2* puncta was quantified. This analysis showed 92.3 ± 0.6% had *Vsx2* puncta, based on three individual sorts. On average, we isolated 200,000 ± 0 *Vsx2*- cells, 96,000 ± 18,600 *Grik1*- cells, and 22,000 ± 1,000 *Grik1*+ cells per biological replicate, with each replicate originating from two retinas. These results indicate that gene-specific SABER-FISH can label dissociated cell populations for isolation by FACS.

To determine whether the isolated populations corresponded to the expected cell types, we reversed the crosslinking and extracted the RNA from these cells. SMART-Seq v.4 cDNA libraries were generated and sequenced on NextSeq 500. Each sample was sequenced to a mean of 15 ± 3 million 75 bp paired-end reads to be able to reliably detect low abundance transcripts. Gene body coverage of mapped reads indicated slight 3′ bias, suggesting mild degradation of RNA, with a projected RNA Integrity Number (RIN) of approximately 4–6 (*Sigurgeirsson et al., 2014*). Unbiased hierarchical clustering showed that samples of the same cell population clustered together (average Pearson correlation between samples within population: r = 0.93) (*Figure 1—figure supplement 1*). The three populations were then analyzed for differential expression (DE). Between each population, the frequency distribution of all *p*-values showed an even distribution of null *p*-values, thus allowing for calculation of adjusted *p*-value using the Benjamini-Hochberg procedure (*Figure 1—figure supplement 1*). Between *Grik1*- and *Grik1*+ populations, we found 1,740 differentially expressed genes (adjusted p-value<0.05) out of 17,649 genes (*Figure 1—figure supplement 1*). The high number of genes detected indicates successful bulk RNA sequencing of low abundance transcripts.

To determine which retinal cell types were enriched in the isolated populations, the DE gene set (adjusted p-value<0.05) was compared to the retinal cell class-specific markers identified by Drop-Seq (see Materials and methods for details of gene set curation) (*Macosko et al., 2015*). The *Vsx2*⁻ population was enriched for markers of all cell classes except for BCs and MG (*Figure 1—figure supplement 2*), as expected from the expression pattern of Vsx2 (*Shekhar et al., 2016*). The *Grik1*⁻ population was enriched for most BC and MG markers, while the *Grik1*⁺ population was enriched for a subset of BC markers (*Figure 1—figure supplement 2*). Accordingly, Gene Set Enrichment Analysis (GSEA) between *Vsx2*⁻ and *Grik1*⁻ populations indicated significant enrichment of rod, cone, AC, HC, and RGC markers in the *Vsx2*⁻ populations and BC and MG markers in the *Grik1*⁻ population (default significance at FDR < 0.25; Enrichment in the *Vsx2*⁻ population: Rod: FDR < 0.001; Cone: FDR < 0.001; AC: FDR < 0.001; HC: FDR = 0.174; RGC: FDR = 0.224; Enrichment in the *Grik1*⁻ population: BC: FDR < 0.001; MG: FDR < 0.001).

To determine which BC subtypes were enriched in the *Grik1*⁻ and *Grik1*⁺ populations, the DE gene set (adjusted p-value<0.05) was cross-referenced to the BC subtype specific markers identified by scRNA sequencing (*Shekhar et al., 2016*). The majority of BC2, BC3A, BC3B, and BC4 markers were enriched in the *Grik1*⁺ population as expected, and all other BC subtype markers were highly expressed in the *Grik1*⁻ population (*Figure 1f*). GSEA between *Grik1*⁻ and *Grik1*⁺ populations confirmed these results (Enrichment in the *Grik1*⁺ population: BC2: FDR < 0.001; BC3A: FDR = 0.005; BC3B: FDR < 0.001; BC4: FDR < 0.001; Enrichment in the *Grik1*⁻ population: BC1B: FDR = 0.132; BC5A: FDR = 0.135; BC5C: FDR = 0.136; BC5D: FDR = 0.172; BC6: FDR = 0.145; BC7: FDR = 0.169; BC8/9: FDR = 0.174; RBC: FDR < 0.001). From the DE analysis, the top 20 most DE genes that were specific to a cell population were identified (*Figure 1—figure supplement 3*). The expression of *Tpbgl*, a previously uncharacterized transcript, was confirmed in *Grik1*⁺ cells by SABER-FISH in retinal tissue sections (*Figure 1—figure supplement 3*). These results indicate that the cell populations isolated and profiled by Probe-Seq correspond to the expected BC subtypes.

We next aimed to determine the relative quality of the transcriptomes obtained by Probe-Seq versus those obtained from freshly dissociated cells. To this end, the Grik1^CRM4-GFP reporter plasmid was electroporated into the developing retina at P2. We previously showed that 72% of GFP⁺ cells were *Grik1*⁺ using this reporter (*Kishi et al., 2019*). At P40, the retinas were harvested, and the electroporated region was dissociated into a single cell suspension. GFP⁺ cells were FACS isolated into Trizol (henceforth called Live cells). Simultaneously, cells from the unelectroporated region from the same retina were used for Probe-Seq of *Grik1*⁺ cells (henceforth called Probe-Seq cells). On average, 836 ± 155 GFP⁺ Live cells (n = 3) and 10,000 ± 0 *Grik1*⁺ Probe-Seq cells (n = 3) were collected for transcriptional analysis. To measure the RNA quality of the Live cells and Probe-Seq cells, 100,000 GFP⁻ Live cells (n = 3) and 100,000 *Grik1*⁻ Probe-Seq cells (n = 3) were collected. Based on the gene body coverage of the 10,000 most highly expressed genes, a slightly higher 3′−5′ bias was observed for the RNA originating in the Probe-Seq population (1.02 ± 0.02) compared to the Live cell population (0.90 ± 0.01), indicating mild degradation of RNA with the Probe-Seq protocol. Based on the gene body coverage graph, the level of degradation for the Probe-Seq population would project to a RIN score of approximately 4–6 (*Sigurgeirsson et al., 2014*) (*Figure 1—figure supplement 4*). Next, the extent to which the gene expression levels correlated between Live and Probe-Seq cells was determined. An analysis of expression levels of all genes between Live and Probe-Seq cells showed an average Pearson correlation of 0.78 ± 0.01 (*Figure 1g*). Since 72% of the GFP⁺ population in the Live cell population was expected to be *Grik1*⁺ cells, the correlation for markers of BC2 – BC4 was expected to be approximately 0.72. Notably, when analyzing only the (BC2 – BC4) subtype-specific markers, the average correlation was 0.66 between Live and Probe-Seq cells, with enrichment of these markers in the Probe-Seq population (*Figure 1g*). Therefore, expression of GFP in *Grik1*⁻ cells using the Grik1^CRM4-GFP reporter plasmid could explain, at least partially, the difference in BC2 – BC4 subtype marker gene expression between the Probe-Seq and Live cell populations. In addition, as predicted by the reporter expression specificity, DE analysis for all genes between these two populations indicated enrichment of rod, cone, and MG markers, as evidenced by high expression of marker genes (i.e. Rho, Opn1mw, and Apoe) in the Live cell population (*Figure 1g*). Despite the imperfect match in the cellular compositions of these two preparations, the strong correlation between the transcriptomes obtained by Probe-Seq and traditional live cell sorting indicates that high quality transcriptomes of specific cell types can be obtained by this method.

We next sought to investigate the parameters for gene-specific probe sets that are important for successful FACS isolation. The ability to resolve a targeted cell population from the total cell population was predicted to be dependent upon the total number of fluorescent probes in that cell, which could be increased by targeting more oligos to each transcript, and/or by targeting more abundant RNA species. To investigate these parameters, three gene-specific probe sets were made for *Grik1*, *Grm6*, and *Neto1*, which exhibit high-to-low levels of gene expression based upon FISH analysis in retina tissue sections (Number of puncta per positive cell: *Grik1:* 37; *Grm6:* 20; *Neto1:* 6.5) (*Figure 1—figure supplement 5*). For each gene-specific probe set, 48, 24, or 12 randomly-chosen oligos were pooled for extension. These were then used for Probe-Seq, and the fluorescent signals from the FACS were analyzed to calculate the 'Stain Index' (SI; see Materials and methods for calculation of SI). This allowed for quantification of the separation of the positive population from the negative population. The SI was found to decrease with the reduced number of oligos and the level of expression of each gene (*Figure 1h*). However, with an SI cutoff of 2, 12 oligos were sufficient for confidence in the separation of gene-positive and negative populations. This was evident only when the events were displayed in a 2-dimensional flow cytometry plot (*Figure 1—figure supplement 5*). Of note, *Neto1*$^+$ BCs constitute ~0.3% of all retinal cells; nonetheless, a clear separation could be observed for this rare population. These results demonstrate that short transcripts, or few oligos can be used successfully to isolate rare cell types using Probe-Seq.

To label multiple cell types, or cellular states, it is often necessary to use a combination of gene-specific probe sets. This has been achieved in tissue sections using 'Exchange-SABER'. Multiple gene-specific probe sets targeting different genes are hybridized simultaneously, and then several different fluorescent oligos are used in a short hybridization reaction to label several different gene-specific probe sets. Imaging is carried out, and then the fluorescent oligos are stripped using conditions that do not strip the gene-specific probe sets. Additional rounds of hybridization using fluorescent oligos targeting different gene-specific probe sets are carried out. This allows fluorescent channels to be reused for detection of different genes in the same cells. Exchange-SABER has enabled the labeling of seven retinal cell classes using three cycles of FISH. To determine whether multiplexing is feasible with Probe-Seq, dissociated mouse retinal cells were incubated with three gene-specific probe sets for *Gad1*, *Vsx2*, and *Grm6*, each with a unique concatemer sequence (*Gad1*.26, *Vsx2*.25, *Grm6*.27). For the first round of flow cytometry, the fluorescent oligos (26.633 nm and 25.488 nm) for detecting *Gad1* and *Vsx2* were applied and assayed by flow cytometry (*Figure 1i*). Then, the remaining cells (i.e. those that were not put through the first round of flow cytometry) were stripped using 50% formamide. The removal of the fluorescent oligos was confirmed by sampling a small number of stripped cells and confirming the lack of signal in a second round of flow cytometry (*Figure 1i*). The fluorescent oligos (27.488 nm) for *Grm6* were then applied to the remaining stripped cells, enabling detection of a distinct population of cells from the same pool of dissociated cells. These results indicate that multiplexed Probe-Seq can allow detection of multiple cell types in the same cell preparation with one overnight hybridization of multiple gene-specific probe sets.

## Probe-Seq enables isolation and RNA sequencing of cell type-specific nuclei from frozen postmortem human tissue

To determine whether Probe-Seq will allow one to access the transcriptomes of the many archived human tissue samples, we tested the method on frozen human retinas. Nuclear preparations were made, as whole cell approaches to frozen cells are not feasible (*Krishnaswami et al., 2016*; *Lake et al., 2016*). The initial test was carried out on frozen mouse retinas. Nuclei were extracted by Dounce homogenization, fixed with 4% PFA, and labeled by a gene-specific probe set for *Grik1* (*Figure 2—figure supplement 1*). *Grik1*$^{HI}$ (not *Grik1*$^{MID}$) and *Grik1*$^-$ populations were isolated by FACS, the nuclear RNA was extracted, and the cDNA was sequenced. The DE gene set (adjusted p-value<0.05) was cross-referenced to BC subtype specific markers. The majority of mouse BC3A, BC3B, and BC4 markers were found to be enriched in the *Grik1*$^+$ population, as expected, and all other BC subtype markers were highly expressed in the *Grik1*$^-$ population (*Figure 2—figure supplement 1*). These results indicate that cell type-specific nuclear RNA from frozen tissue can be isolated by Probe-Seq.

Fresh-frozen human retinas (age range: 40–60; see Materials and methods for full description of samples) were then tested for isolation and profiling of human BC subtypes using a probe set for

*GRM6*. This gene is expressed in cone ON bipolar cells and rod bipolar cells (RBC) in the mouse and human retina (*Shekhar et al., 2016*; *Cowan et al., 2019*) (*Figure 2a*). To test the *GRM6* probe set in human retinas, it was first applied to a fixed human tissue section, where signal was observed in the expected pattern, in a subset of cells in the inner nuclear layer, where BCs reside (*Figure 2b*). Nuclei were extracted from frozen human peripheral retinas, fixed, and incubated with the *GRM6* probe

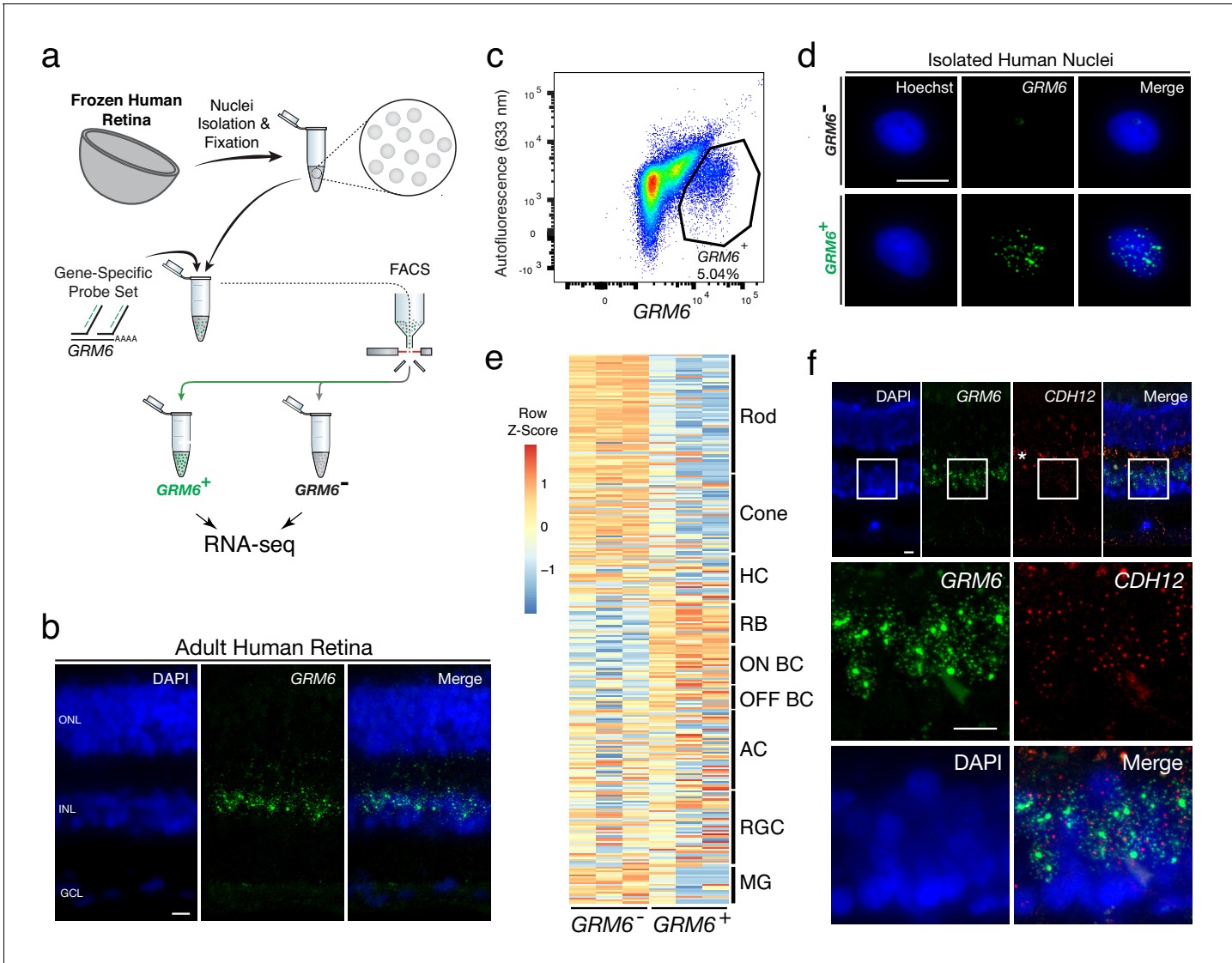

**Figure 2.** Transcriptional profiling of nuclear RNA isolated from specific BC subtypes from frozen human retina. (**a**) Schematic of Probe-Seq for the fresh frozen adult human retina. Single nuclei were prepared and then fixed. Nuclei were incubated with a gene-specific probe set for *GRM6* and then incubated with fluorescent oligos. *GRM6*⁺ and *GRM6*⁻ populations were isolated by FACS for downstream RNA sequencing. (**b**) Image of an adult human retina section probed with a SABER *GRM6* probe set. (**c**) FACS plot of all single nuclei with *GRM6* fluorescence on the x-axis and autofluorescence (633 nm) on the y-axis. (**d**) Images of isolated nuclei processed using SABER-FISH for *GRM6*. (**e**) A heatmap representing relative expression levels of human retinal cell type markers previously identified by scRNA sequencing that are differentially expressed (adjusted p-value<0.05) between *GRM6*⁻, and *GRM6*⁺ populations. (**f**) High and low magnification images of a human retinal section after the SABER-FISH protocol for *CDH12*, a highly-enriched transcript in the *GRM6*⁺ population. Asterisk indicates high autofluorescence in the OPL. HC, Horizontal Cell; RGC, Retinal Ganglion Cell; AC, Amacrine Cell; ON BC, ON Bipolar Cell; RBC, Rod Bipolar Cell; OFF BC, OFF Bipolar Cell; MG, Müller Glia; ONL, Outer Nuclear Layer; INL, Inner Nuclear Layer; GCL, Ganglion Cell Layer. OPL, Outer Plexiform Layer. Scale bars: 10 µm (**b, d, f**).

The online version of this article includes the following figure supplement(s) for figure 2:

**Figure supplement 1.** Isolation and transcriptional profiling of specific BC subtypes from the frozen mouse retina.

**Figure supplement 2.** Quality control of human nuclear Probe-Seq RNA sequencing.

set. The *GRM6*⁻ and *GRM6*⁺ nuclei were then isolated by FACS after application of the fluorescent oligos (*Figure 2c*). On average, 43,000 ± 35,500 *GRM*⁻ nuclei and 1,800 ± 781 GRM6⁺ nuclei were isolated from an approximately 5 mm x 5 mm square of the retina per biological replicate (*Figure 2d*). SMART-Seq v.4 cDNA libraries were sequenced on NextSeq 500, with each sample sequenced to a mean depth of 18 ± 3 million 75 bp paired-end reads. Quality control of the read mapping and DE analysis indicated successful RNA sequencing and DE analysis (*Figure 2—figure supplement 2*). Upon filtering out genes with zero counts in more than four samples, 1956 out of 9,619 genes were differentially expressed (adjusted p-value<0.05).

The DE gene set (adjusted p-value<0.05) was compared to human retinal cell type-specific markers identified by scRNA sequencing (*Cowan et al., 2019*). An enrichment of markers for RBCs and ON BCs were found in the *GRM6*⁺ population (*Figure 2e*). GSEA between *GRM6*⁻ and *GRM6*⁺ confirmed these results (Enrichment in *GRM6*⁺ population: RBC: FDR = 0.003; ON BC-1: FDR = 0.181; ON BC-2: FDR = 0.126), indicating that the expected human retinal populations were accurately isolated. The expression of *CDH12*, a transcript highly enriched in the *GRM6*⁺ population, but previously not reported to be a marker of these subtypes, was confirmed using SABER-FISH on fixed adult human retina sections (*Figure 2f*). These results show that nuclear transcripts isolated from frozen tissue by Probe-Seq are sufficient for transcriptional profiling.

## Isolation and transcriptional profiling of intestinal stem cells from the *Drosophila* midgut

To determine whether Probe-Seq can be successfully applied to non-CNS cells, and to cells from invertebrates, we applied the method to the midgut of *Drosophila melanogaster*. The adult *Drosophila* midgut is composed of four major cell types – enterocytes (EC), enteroendocrine cells (EE), enteroblasts (EB), and intestinal stem cells (ISC), though recent profiling studies have revealed heterogeneity among ECs and EEs (*Dutta et al., 2015*; *Hung et al., 2018*). We aimed to isolate ISCs and EBs using a gene-specific probe set for *escargot* (*esg*), a well-characterized marker for these cell types (*Figure 3a*). As SABER-FISH had not yet been tested on Drosophila tissue, this method was first tested on wholemounts of the *Drosophila* gut. SABER-FISH signal was observed in the appropriate pattern, in a subset of midgut cells (*Figure 3b*). To perform Probe-Seq, 35–40 *Drosophila* midguts per biological replicate were dissociated, fixed in 4% PFA, and incubated with the *esg* probe set. FACS was then used to sort *esg*⁻ and *esg*⁺ populations (*Figure 3c - d*). As only the 2N single cells were sorted, proliferating (S – M phase) ISCs and polyploid ECs were likely excluded. On average, 1,400 ± 208 *esg*⁻ cells and 1,000 ± 404 *esg*⁺ cells per biological replicate were isolated. SMART-Seq v.4 cDNA libraries were sequenced on NextSeq 500 to a mean of 16 ± 4 million 75 bp paired-end reads. Quality control of the read mapping and DE analysis indicated successful RNA sequencing and DE analysis (*Figure 3—figure supplement 1*). Upon filtering out genes with zero counts in more than three samples, 405 out of 1,596 genes were differentially expressed (adjusted p-value<0.05).

The DE gene set (adjusted p-value<0.05) from Probe-Seq was compared to cell type-specific markers identified by scRNA sequencing (*Hung et al., 2018*). An enrichment of ISC/EB markers was observed in the *esg*⁺ population isolated using Probe-Seq, while markers of all other cell types were enriched in the *esg*⁻ population (*Figure 3e*). GSEA between *esg*⁺ and *esg*⁻ populations isolated using Probe-Seq indicated significant enrichment of ISC/EB markers in the *esg*⁺ population and all other cell type markers in the *esg*⁻ population (Enrichment in *esg*⁺ population: ISC/EB: FDR < 0.001; Enrichment in *esg*⁻ population: LFC: FDR < 0.001; aEC1: FDR < 0.001; aEC2: FDR < 0.001; aEC3: FDR < 0.001; pEC3: FDR < 0.001; AstA-EE: FDR < 0.001; mEC3: FDR < 0.001; Copper-Iron: FDR = 0.001; pEC2: FDR = 0.001; AstC: FDR = 0.004; aEC4: FDR = 0.003; pEC1: FDR = 0.004; dEC: FDR = 0.013; NPF-EE: FDR = 0.019). Using SABER-FISH on wholemounts of midguts, the co-localization of *esg* and *Sox100B*, a transcript significantly enriched in the *esg*⁺ population, was validated (*Figure 3f*). Additionally, the Probe-Seq DE gene set (adjusted p-value<0.05) was cross-referenced with the ISC/EB and EC markers defined by DamID profiling of the adult *Drosophila* gut (*Doupé et al., 2018*). From this analysis, the majority of ISC/EB and EC markers was seen to be enriched in *esg*⁺ and *esg*⁻ populations, respectively (*Figure 3—figure supplement 2*). These results demonstrate that Probe-Seq enables the isolation and transcriptional profiling of specific cell types from invertebrate tissue and from non-CNS tissue.

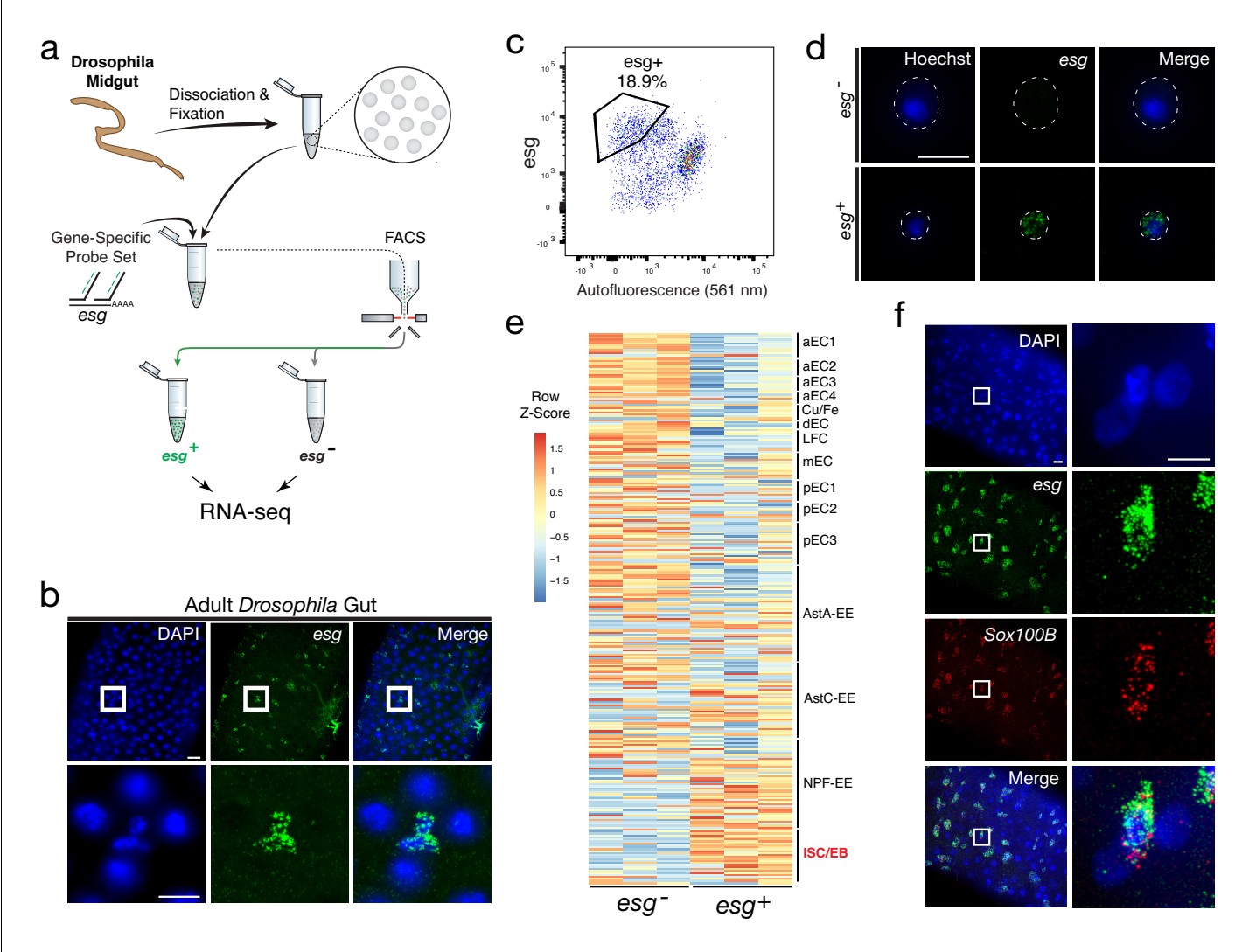

**Figure 3.** Isolation and transcriptional profiling of ISC/EBs from the adult *Drosophila* midgut. (**a**) Schematic of Probe-Seq for the adult *Drosophila* midgut. Midguts from 7 to 10 day old female flies were dissociated into single cells and fixed. Cells were incubated with a gene-specific probe set for *esg* and subsequently incubated with fluorescent oligos. *esg*+ and *esg*- populations were isolated by FACS for downstream RNA sequencing. (**b**) Image of a wholemount adult *Drosophila* midgut following the SABER-FISH protocol using an *esg* probe set. (**c**) FACS plot of all single cells with *esg* fluorescence on the y-axis and autofluorescence (561 nm) on the x-axis. (**d**) Images of isolated midgut cells processed using SABER-FISH for *esg* before FACS. The white dotted lines demarcate cell outlines. (**e**) A heatmap representing relative expression levels of differentially expressed (adjusted p-value<0.05) genes for *Drosophila* gut cell type markers previously identified by scRNA sequencing, between *esg*-, and *esg*+ populations. *esg* is expressed in the ISC/EB population (highlighted in red). (**f**) Images of a *Drosophila* midgut wholemount after the SABER-FISH protocol for an ISC/EB marker, *Sox100B*, a highly-enriched transcript in the *esg*+ population. EC, Enterocyte; ISC, Intestinal Stem Cell; EB, Enteroblast; EE, Enteroendocrine Cell; LFC, Large Flat Cell; Cu/Fe, Copper/Iron Cells. Scale bars: 10 µm (d, f, right panels); 20 µm (b, upper panels f, left panels).

The online version of this article includes the following figure supplement(s) for figure 3:

**Figure supplement 1.** Quality control of *Drosophila* Probe-Seq RNA sequencing.

**Figure supplement 2.** Heatmap of ISC/EB and EC markers based on DamID transcriptional profiling.

## Transcriptome profiling of the central chick retina reveals unique transcripts expressed in cells that give rise to the high acuity area

The central chicken retina contains a region thought to endow high acuity vision, given its cellular composition and arrangement of cells (*da Silva and Cepko, 2017*). It comprises a small and discrete area that is devoid of rod photoreceptors and enriched in cone photoreceptors, with a high density of RGCs, the output neurons of the retina (*da Silva and Cepko, 2017*; *Bruhn and Cepko, 1996*).

These features are shared with the high acuity areas of other species, including human. Although we have shown that *FGF8, CYP26C1,* and *CYP26A1* are highly enriched in this area at embryonic day 6 (E6) (*da Silva and Cepko, 2017*), the other molecular determinants that may play a role in high acuity area development are unknown. Probe-Seq was thus used to isolate and sequence *FGF8⁺* cells (*Figure 4a*). Hamburger-Hamilton stage 28 (HH28) chick retinas were dissociated into single cells, fixed, and probed for the *FGF8* transcript, which is expressed in the high acuity area at E6

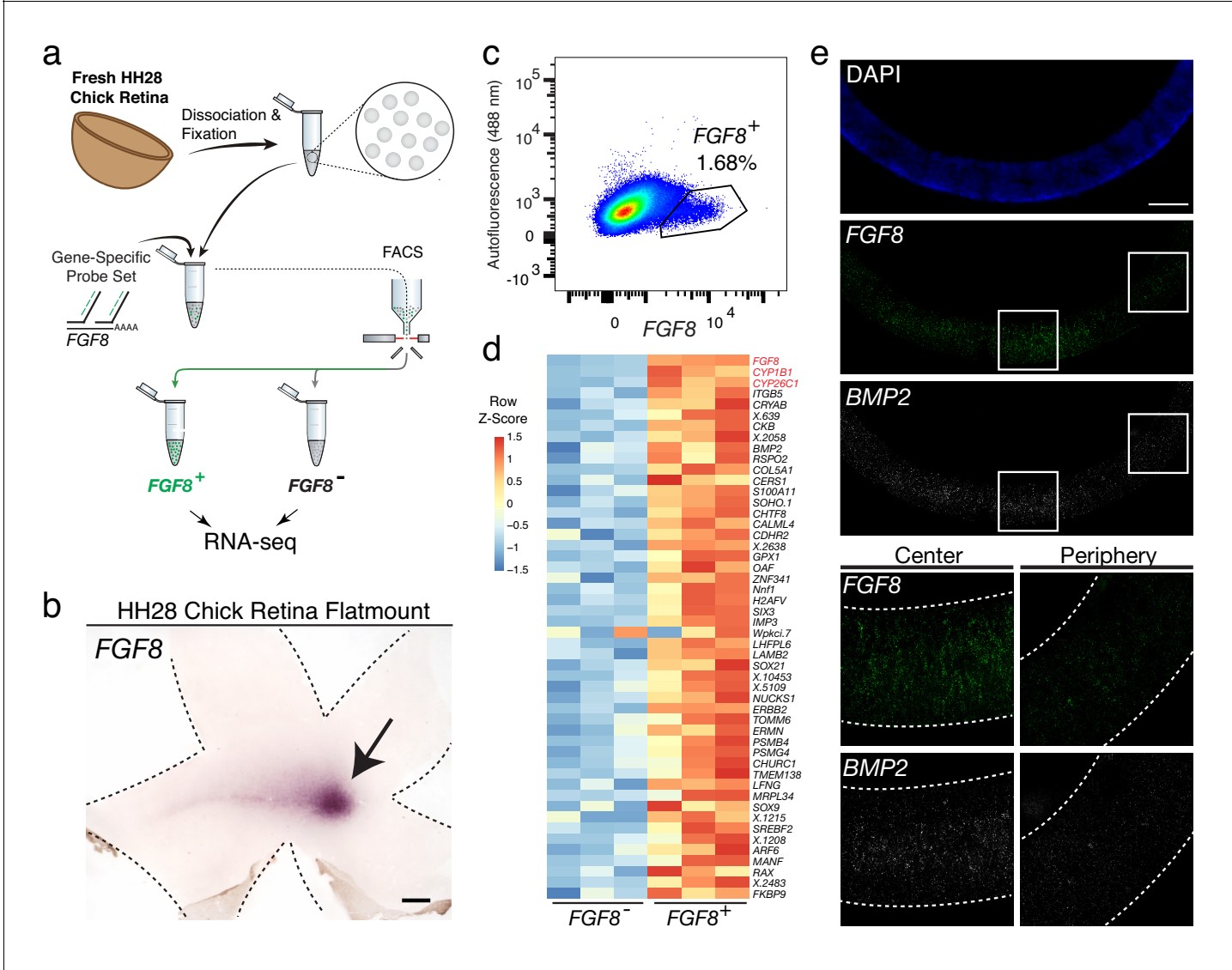

**Figure 4.** Probe-Seq identifies the transcriptional landscape of chick central retinal progenitor cells that give rise to the high acuity area. (a) Schematic of Probe-Seq for the developing HH28 chick retina. The chick retina was dissociated into single cells and fixed. Cells were incubated with a gene-specific probe set for *FGF8* and subsequently incubated with fluorescent oligos. *FGF8⁺* and *FGF8⁻* populations were isolated by FACS for downstream RNA sequencing. (b) In situ hybridization of *FGF8* on a HH28 chick retina flatmount. Arrow indicates region of *FGF8* expression. Dotted lines demarcate retina outline. (c) FACS plot of all single cells with *FGF8* fluorescence on the x-axis and empty autofluorescence (488 nm) on the y-axis. (d) A heatmap of unbiased top 50 genes that were enriched in the *FGF8⁺* population compared to the *FGF8⁻* population. (e) Images of a section spanning the central HH28 chick retina after the SABER-FISH protocol for *FGF8* and *BMP2*, a transcript highly enriched in the *FGF8⁺* population, Scale bars: 500 μm (b); 50 μm (e).

The online version of this article includes the following figure supplement(s) for figure 4:

**Figure supplement 1.** Uniform distribution of mapped reads from embryonic chick Probe-Seq for *FGF8*.

**Figure supplement 2.** Quality control of developing chick retina Probe-Seq RNA sequencing.

**Figure supplement 3.** Early differentiation markers are enriched in the *FGF8⁻* population.

(*Figure 4b*). On average, 7,000 ± 6,950 *FGF8*$^+$ cells and 189,000 ± 19,000 *FGF8*$^-$ cells were FACS isolated into individual populations (*Figure 4c*). The cDNA from each population (n = 3) was sequenced to a mean depth of 18 ± 7 million 75 bp paired-end reads on NextSeq 500. Interestingly, gene body coverage of mapped reads showed more uniform distribution compared to that of adult mouse retina, suggesting reduced RNA degradation. (*Figure 4—figure supplement 1*). Quality control of the read mapping and DE analysis indicated successful RNA sequencing and DE analysis (*Figure 4—figure supplement 2*). Between *FGF8*$^-$ and *FGF8*$^+$ populations, we found 1,924 DE genes out of 12,053 genes.

Among the top 50 most enriched DE transcripts in the *FGF8*$^+$ population were *FGF8*, *CYP1B1*, and *CYP26C1* (*Figure 4d*). The latter two transcripts are components of the retinoic acid signaling pathway, and were previously shown to be highly enriched in the central retina where *FGF8* is expressed (*da Silva and Cepko, 2017*). Previously, *FGF8* expression was shown to be largely confined to the area where progenitor cells reside. However, it was unclear whether it was also expressed in differentiated cells (*da Silva and Cepko, 2017*). DE analysis of the *FGF8*$^-$ and *FGF8*$^+$ populations revealed enrichment of early differentiation markers of RGCs (i.e. *NEFL*) and photoreceptors (i.e. *NEUROD1*) in the *FGF8*$^-$ population, confirming that *FGF8* is mostly expressed in central progenitor cells (*Figure 4—figure supplement 3*). As we wished to validate the DE genes using FISH on sections, and SABER-FISH had not yet been tested on chick tissue, we first tested the method using the *FGF8* probe set on chick tissue sections (*Figure 4e*). Robust and specific FISH signal was seen in the appropriate pattern for *FGF8*. An additional SABER-FISH probe set for *BMP2*, a transcript enriched in the *FGF8*$^+$ population, was then used on developing HH28 chick central retinal sections. *BMP2* was found to be highly enriched within the *FGF8*$^+$ population, that is, expression was confined to a discrete central retina where *FGF8* was expressed (*Figure 4e*). These results indicate that Probe-Seq of the developing chick retina using *FGF8* as a molecular handle can reveal the transcriptional profile of the progenitor cells that will comprise the chick high acuity area.

## Discussion

Studies of model organisms have allowed the dissection of molecular mechanisms that underlie a variety of biological processes. However, each organism across the evolutionary tree possesses unique traits, and understanding these traits will greatly enrich our understanding of biological processes. Non-model organisms can now be investigated at the genetic level, due to advances in DNA sequencing, transcriptional profiling, and genome modification methods. Despite progress, challenges remain to achieve greater depth in the characterization of the transcriptomes of rarer cell types within heterogeneous tissues. Even in model organisms, deep transcriptional profiling of specific cell types remains difficult if specific *cis*-regulatory elements are unavailable. To overcome these challenges, we developed Probe-Seq.

Probe-Seq uses a FISH method based upon SABER probes to hybridize gene-specific probe sets to RNAs of interest (*Kishi et al., 2019*). This method provides amplified fluorescent detection of RNA molecules and can be spectrally or serially multiplexed to mark cell populations based on combinatorial RNA expression profiles. Previously-identified markers can thus be targeted by gene-specific probe sets to isolate specific cell types by FACS. Subsequently, deep RNA sequencing can be carried out on the sorted population to generate cell type-specific transcriptome profiles. Due to the reliance on RNA for cell sorting, rather than protein, this method is applicable across organisms. Other RNA-based methods to label specific cell types for downstream sequencing have not been tested with tissue samples, and/or require cell encapsulation in a microfluidic device (*Klemm et al., 2014*; *Eastburn et al., 2014*; *Pellegrino et al., 2016*).

Probe-Seq allowed the isolation and profiling of RNA from fresh mouse, frozen human, and fresh chick retinas, as well as gut cells from *Drosophila melanogaster*. Aside from the different dissociation protocols, Probe-Seq does not require species- or tissue-specific alterations. To profile multiple cellular subtypes, multiplexed Probe-Seq allows for iterative labeling, sorting, and re-labeling with one overnight hybridization of multiple gene-specific probe sets. This strategy enables separation of FACS-isolated, broad populations into finer sub-populations. The Probe-Seq method is also cost and time effective, with less than 6 hr of hands-on time, including FISH, FACS, and library preparation. Excluding the wait time for oligo delivery, libraries ready for sequencing can be made by Probe-Seq in less than three days. Per sample, we estimate the cost to be less than $200, from start

to finish, achieving ~15 million paired-end reads. The Probe-Seq protocol may be further optimized to maximize utility. For example, the protocol may be further modified to use other single molecule FISH methods such as clampFISH (*Rouhanifard et al., 2019*) or RNAscope (*Wang et al., 2012*) rather than SABER-FISH, as these methods have their respective strengths and weaknesses. Additionally, the protocol may be made for efficient for cell capture, for example a cell strainer can be used for the wash steps to minimize cell loss from centrifugation. Further development for scRNA sequencing after cell type enrichment using Probe-Seq may also be possible. For this, however, adaptation of Probe-Seq for the reversal of crosslinks for scRNA sequencing (*Alles et al., 2017*; *Attar et al., 2018*; *Chen et al., 2018*) will likely be necessary.

# Materials and methods

## Key resources table

| Reagent type (species) or resource | Designation | Source or reference | Identifiers | Additional information |
|---|---|---|---|---|
| Gene (*Mus musculus*) | Vsx2 | | Ensembl: ENSMUSG00000021239 | |
| Gene (*Mus musculus*) | Grik1 | | Ensembl: ENSMUSG00000022935 | |
| Gene (*Mus musculus*) | Tpbgl | | Ensembl: ENSMUSG00000096606 | |
| Gene (*Mus musculus*) | Grm6 | | Ensembl: ENSMUSG00000000617 | |
| Gene (*Mus musculus*) | Neto1 | | Ensembl: ENSMUSG00000050321 | |
| Gene (*Mus musculus*) | Gad1 | | Ensembl: ENSMUSG00000070880 | |
| Gene (*Homo sapiens*) | GRM6 | | Ensembl: ENSG00000113262 | |
| Gene (*Homo sapiens*) | CDH12 | | Ensembl: ENSG00000154162 | |
| Gene (*Drosophila melanogaster*) | esg | | FlyBaseID: FBgn0001981 | |
| Gene (*Drosophila melanogaster*) | Sox100B | | FlyBaseID: FBgn0024288 | |
| Gene (*Gallus gallus*) | FGF8 | | Ensembl: ENSGALG00000007706 | |
| Gene (*Gallus gallus*) | BMP2 | | Ensembl: ENSGALG00000029301 | |
| Strain, strain background (*Mus musculus*) | CD1 | Charles River Laboratories | 022 | |
| Strain, strain background (*Drosophila melanogaster*) | Oregon-R | | | |
| Strain, strain background (*Gallus gallus*) | White Leghorn | Charles River Laboratories | | |
| Sequence-based reagent | Variable oligos | This Paper | See Supplementary Table for sequences | |
| Commercial assay or kit | RecoverAll Total Nuclear Isolation Kit | Thermo Fisher Scientific | AM1975 | |
| Commercial assay or kit | SMART-Seq v.4 Ultra Low Input RNA kit | Takara Bio | 634890 | |
| Commercial assay or kit | Nextera XT DNA Library Prep Kit | Illumina | FC1311096 | |

## Mouse retina samples

All animals were handled according to protocols approved by the Institutional Animal Care and Use Committee (IACUC) of Harvard University. For fresh samples, retinas of adult CD1 mice (>P30) from Charles River Laboratories were dissected. For frozen samples, retinas of adult CD1 mice (>P30) were dissected and frozen in a slurry of isopentane and dry ice and kept at −80℃.

## Human retina samples

Frozen eyes were obtained from Restore Life USA (Elizabethton, TN) through TissueForResearch. Patient DRLU041818C was a 53-year-old female with no clinical eye diagnosis and the postmortem interval was 9 hr. Patient DRLU051918A was a 43-year-old female with no clinical eye diagnosis and the postmortem interval was 5 hr. Patient DRLU031318A was a 47-year-old female with no clinical eye diagnosis and the postmortem interval was 7 hr. This IRB protocol (IRB17-1781) was determined to be not-human subject research by the Harvard University Area Committee on the Use of Human Subjects.

## Chick retina samples

Fertilized White Leghorn eggs from Charles River Laboratories were incubated at 38℃ with 40% humidity. Embryos were staged according to Hamburger and Hamilton up to HH28 (*Hamburger and Hamilton, 1951*).

## *Drosophila melanogaster* midgut samples

Tissues were dissected from female 7–10 day-old adult Oregon-R *Drosophila melanogaster*. Flies were reared on standard cornmeal/agar medium in 12:12 light:dark cycles at 25℃.

## Dissociation of mouse and chick retinas

Mouse or chick retinas were dissected away from other ocular tissues in Hank's Balanced Salt Solution (Thermo Fisher Scientific, cat. #14025092) or PBS. The retina was then transferred to a microcentrifuge tube and incubated for 7 min at 37℃ with an activated papain dissociation solution (87.5 mM HEPES pH 7.0 (Thermo Fisher Scientific, cat. #15630080), 2.5 mM L-Cysteine (MilliporeSigma, cat. # 168149), 0.5 mM EDTA pH 8.0 (Thermo Fisher Scientific, cat. #AM9260G), 10 µL Papain Suspension (Worthington, cat. #LS0003126), 19.6 µL UltraPure Nuclease-Free Water (Thermo Fisher Scientific, cat. #10977023), HBSS up to 400 µL, activated by a 15 min incubation at 37℃). The retina was then centrifuged at 600 xg for 3 min. The supernatant was removed, and 1 mL of HBSS/10% FBS (Thermo Fisher Scientific, cat. #10437028) was added without agitation to the pellet. The pellet was centrifuged at 600 xg for 3 min. The supernatant was removed, and 600 µL of trituration buffer (DMEM (Thermo Fisher Scientific, cat. #11995065), 0.4% (wt/vol) Bovine Serum Albumin (MilliporeSigma cat. #A9418)) was added. The pellet was dissociated by trituration at room temperature (RT) using a P1000 pipette up to 20 times or until the solution was homogenous.

## Dissociation of *Drosophila* midgut

35–40 *Drosophila* midguts were dissected in PBS and transferred to 1% BSA/PBS solution. The midguts were incubated in 400 µL of Elastase/PBS solution (1 mg/mL, MilliporeSigma cat. #E0258) for 30 min to 1 hr at RT, with trituration with a P1000 pipette every 15 min. 1 mL of 1% BSA/PBS was then added. This solution was overlaid on top of Optiprep/PBS (MilliporeSigma, cat. #D1556) solution with a density of 1.12 g/mL in a 5 mL polypropylene tube (Thermo Fisher Scientific, cat. #1495911A). The solution was centrifuged at 800 xg at RT for 20 min. The top layer with viable cells was collected for further processing.

## Mouse and human frozen nuclei isolation

Upon thawing, tissue was immediately incubated in 1% PFA (with 1 µL mL$^{-1}$ RNasin Plus (Promega, cat. #N2611)) for 5 min at 4℃. Nuclei were prepared by Dounce homogenizing in Homogenization Buffer (250 mM sucrose, 25 mM KCl, 5 mM MgCl$_2$, 10 mM Tris buffer, pH 8.0, 1 µM DTT, 1x Protease Inhibitor (Promega, cat. #6521), Hoechst 33342 10 µg mL$^{-1}$ (Thermo Fisher Scientific, cat. #H3570), 0.1% Triton X-100, 1 µL mL$^{-1}$ RNasin Plus). Sample was then overlaid on top of a 20%

sucrose solution (25 mM KCl, 5 mM MgCl$_2$, 10 mM Tris buffer, pH 8.0) and spun at 500 xg for 12 min at 4℃.

## Probe-Seq for whole cells and nuclei

Step-by-step protocol is available at https://doi.org/10.17504/protocols.io.6j3hcqn.

For all solutions, 1 µL mL$^{-1}$ RNasin Plus was added 10 min before use. If the cells or nuclei were not already pelleted, the suspended cells/nuclei (henceforth called cells) were centrifuged at 600 xg for 5 min at 4℃. The cells were then resuspended in 1 mL of 4% PFA (Electron Microscopy Sciences, cat. #15714S, diluted in PBS) and incubated at 4℃ for 15 min with rocking. The cells were centrifuged at 2000 xg for 5 min at 4℃. Except in the case of nuclei, the supernatant was removed, and the cells were resuspended in 1 mL of Permeabilization Buffer (Hoechst 33342 10 µg mL$^{-1}$, 0.1% Trixon X-100, PBS up to 1 mL) and incubated for 10 min at 4℃ with rocking. For both cells and nuclei, the cells were next centrifuged at 2000 xg for 5 min at 4℃. The supernatant was removed, and the cells were resuspended in 500 µL of pre-warmed (43℃) 40% wash Hybridization solution (wHyb; 2x SSC (Thermo Fisher Scientific, cat. #15557044), 40% deionized formamide (Millipore-Sigma, cat. #S4117), diluted in UltraPure Water). Compared to the original SABER protocol (*Kishi et al., 2019*), the Tween-20 was removed as we found that it causes cell clumping. The cells were incubated for at least 30 min at 43℃. After this step, the cell pellet became transparent. The cells were centrifuged at 2000 xg for 5 min at RT, and the supernatant was carefully removed, leaving ~100 µL of supernatant. The cells were then resuspended in 100 µL of pre-warmed (43℃) probe mix (1 µg of probe per gene, 96 µL of Hyb1 solution (2.5x SSC, 50% deionized formamide, 12.5% Dextran Sulfate (MilliporeSigma cat. #D8906)), diluted up to 120 µL with UltraPure Water) and incubated overnight (16–24 hr) at 43℃.

500 µL of pre-warmed (43℃) 40% wHyb was added to the cells and centrifuged at 2000 xg for 5 min at RT. The supernatant was removed, and the cells were resuspended in 500 µL of pre-warmed (43℃) 40% wHyb. The cells were incubated for 15 min at 43℃. The cells were then centrifuged at 2000 xg for 5 min at RT, and the supernatant was removed. 1 mL of pre-warmed (43℃) 2x SSC solution was added, the cells were resuspended, and incubated for 5 min at 43℃. The cells were centrifuged at 2000 xg for 5 min at RT, and the supernatant was removed. Cells were then resuspended in 500 µL of pre-warmed (37℃) PBS. The cells were centrifuged at 2000 xg for 5 min at RT, and the supernatant was removed. The cells were resuspended in 100 µL of fluorescent oligo mix (100 µL of PBS, 2 µL of each 10 µM fluorescent oligo) and incubated for 10 min at 37℃. After incubation, 500 µL of pre-warmed (37℃) PBS was added and the cells were centrifuged at 2000 xg for 5 min at RT. The supernatant was removed, and the cells were resuspended in 500 µL of pre-warmed (37℃) PBS. The cells were incubated for 5 min at 37℃. The cells were centrifuged at 2000 xg for 5 min at RT, the supernatant was removed, and the cells were resuspended in 500–1000 µL of PBS, depending on cell concentration.

## FACS isolation of specific cell types

The suspended labeled cells were kept on ice before FACS. Immediately before FACS, the cells were filtered through a 35 µm filter (Thermo Fisher Scientific, cat. #352235) for mouse, chick, and human retina cells/nuclei or a 70 µm filter (Thermo Fisher Scientific, cat. #352350) for *Drosophila* cells. FACSAria (BD Biosciences) with 488 nm, 561 nm or 594 nm, and 633 nm lasers was used for the sorts. 2N single cells were gated based on the Hoechst histogram. For the *Drosophila* gut, debris was gated out first by FSC/SSC plot because the high number of debris events masked the Hoechst$^+$ peaks. Out of the single cells, a 2-dimensional plot (with one axis being the fluorescent channel of interest and another axis that is empty) was used to plot the negative and positive populations. The events that ran along the diagonal in this plot were considered negative, and the positive events were either left- or right-shifted (depending on axes) compared to the diagonal events. The empty channel was used to determine autofluorescence, and the events that displayed high intensity for both the channel of interest and the empty channel were deemed negative as they may be events with high autofluorescence in all channels. For some samples, the number of sorted cells was capped, as indicated by a standard deviation of 0. Different populations were sorted into microcentrifuge tubes with 500 µL of PBS and kept on ice after FACS. The protocol was later modified so

that the cells were sorted into 500 µL of 1% BSA/PBS, as this significantly improved cell pelleting. The data obtained for this study did not use 1% BSA/PBS.

## RNA isolation and library preparation

The sorted cells were transferred to a 5 mL polypropylene tube and centrifuged at 3000 xg for 7 min at RT. The supernatant was removed as much as possible, the cells were resuspended in 100 µL Digestion Mix (RecoverAll Total Nuclear Isolation Kit (Thermo Fisher Scientific, cat. #AM1975) 100 µL of Digestion Buffer, 4 µL of protease), and incubated for 3 hr at 50℃, which differs from the manufacturer's protocol. The downstream steps were according to the manufacturer's protocol. The volume of ethanol/additive mix in the kit was adjusted based on the total volume (100 µL of Digestion Mix and remaining volume after cell pelleting). The libraries for RNA sequencing were generated using the SMART-Seq v.4 Ultra Low Input RNA kit (Takara Bio, cat. #634890) and Nextera XT DNA Library Prep Kit (Illumina, cat. #FC1311096) according to the manufacturer's protocol. The number of cycles for SMART-Seq v.4 protocol was as follows: Mouse *Vsx2/Grik1*: 13 cycles; Chick *FGF8*: 16 cycles; Human *GRM6:* 16 cycles; *Drosophila esg*: 17 cycles. 150 pg of total cDNA was used as the input for Nextera XT after SMART-Seq v.4, and 12 cycles were used except for *Drosophila* samples for which 14 cycles were necessary. The cDNA library fragment size was determined by the BioAnalyzer 2100 HS DNA Assay (Agilent, cat. #50674626). The libraries were sequenced as 75 bp paired-end reads on NextSeq 500 (Illumina).

## RNA-Seq data analysis

Quality control of RNA-seq reads were performed using fastqc version 0.10.1 (https://www.bioinformatics.babraham.ac.uk/projects/fastqc/). RNA-seq reads were clipped and mapped onto the either the mouse genome (Ensembl GRCm38.90), human genome (Ensembl GRCh38.94), chick genome (Ensembl GRCg6a.96), or *Drosophila* genome (BDGP6.22) using STAR version 2.5.2b (*Aken et al., 2017*; *Dobin et al., 2013*). Parameters used were as follows: `-runThreadN 6 -readFilesCommand zcat -outSAMtype BAM SortedByCoordinate -outSAMunmapped Within -outSAMattributes Standard -clip3pAdapterSeq -quantMode TranscriptomeSAM GeneCounts`.

Alignment quality control was performed using Qualimap version 2.2.2 (*Okonechnikov et al., 2016*). The 3′−5′ bias was determined using the 10,000 most highly expressed genes and 100 bp as the 3′ and 5′ ends and taking the reciprocal of the 5′−3′ bias. Read counts were produced by HT-seq version 0.9.1 (*Anders et al., 2015*). Parameters used were as follows: `-i gene_name -s no`.

The resulting matrix of read counts were analyzed for differential expression by DESeq2 version 3.9 (*Love et al., 2014*). For the DE analysis of human and *Drosophila* samples, any genes with more than 4 and 3 samples with zero reads, respectively, were discarded. The R scripts used for differential expression analysis are available in *Source code 1*.

## Gene set curation

Unique marker genes that define different cell types in different tissue types were curated in an unbiased manner. For the mouse retina, marker genes of major cell types were identified from scRNA sequencing (*Macosko et al., 2015*). Genes that were found in more than one cluster were removed from the analysis to obtain unique cluster-specific markers. Rod-specific genes were highly represented in all clusters; thus, they were considered non-unique by this analysis. Therefore, the top 20 rod-specific genes were manually added after non-unique genes were removed. For mouse BCs, marker genes of BC subtypes with high confidence were identified by scRNA sequencing (*Shekhar et al., 2016*). Genes that were found in more than one cluster was removed from the analysis. For the human retina, marker genes of major cell types were identified by scRNA sequencing (*Cowan et al., 2019*). Marker genes that were expressed in <90% of cells in the cluster were removed for analysis. For the *Drosophila* gut, marker genes of major cell types were identified from DamID transcriptional profiling and scRNA sequencing (*Hung et al., 2018*; *Doupé et al., 2018*). For the *Drosophila* scRNA sequencing dataset, marker genes with <0.75 avgLogFC were discarded, and only the major cell types were used. For the DamID dataset, a cutoff of FDR < 0.01 was used for marker genes that were specifically expressed between ISC/EBs and ECs.

## Gene set enrichment analysis

GSEAPreranked analysis was performed using GSEA v3.0 (*Subramanian et al., 2005*). Curated gene sets described above were used to define various cell types. Parameters used were as follows: Number of permutations: 1000; Enrichment statistic: classic; the ranked file was generated using log2-FoldChange generated by DESeq2. To determine significance, we used the default FDR < 0.25 for all gene sets.

## SABER gene-specific probe set synthesis

SABER gene-specific probe sets were synthesized using the original protocol (*Kishi et al., 2019*). The gene of interest was searched in the UCSC Genome Browser. Then, the BED files for genes of interest were generated through the UCSC Table Browser with the following parameters: group: Genes and Gene Predictions; track: NCBI RefSeq; table: UCSC RefSeq (refGene); region: position; output format: BED; Create one BED record per: Exons. If multiple isoforms were present in the BED output file, all but one was removed manually. Genome-wide probe sets for mouse, human, chick, and *Drosophila* were downloaded from (https://oligopaints.hms.harvard.edu/genome-files) with Balance setting. The oligo sequences were generated using intersectBed (bedtools 2.27.1) between the BED output file and the genome-wide chromosome-specific BED file with $-f\ 1$. If the BED file sequences were on the + strand, the reverse complement probe set was generated using OligoMiner's probeRC.py script (https://github.com/brianbeliveau/OligoMiner). For each oligo sequence, hairpin primer sequences were added following a 'TTT' linker. The oligos were ordered from IDT with the following specifications in a 96-well format: 10 nmole, resuspended in IDTE pH 7.5, V-Bottom Plate, and normalized to 80 μM. The oligos were then combined into one pool using a multi-channel pipette. The oligo sequences for every gene-specific probe set used in this study are provided in Supplementary Files.

The pooled oligos were extended by a Primer Exchange Reaction concatemerization reaction (1X PBS, 10 mM $MgSO_4$, dNTP (0.3 mM of A, C, and T), 0.1 μM Clean.G, 0.5 μL Bst Polymerase (McLab, cat. #BPL-300), 0.5 μM hairpin, 1 μM oligo pool). The reaction was incubated without the oligo pool for 15 min at 37°C. Then, the oligo pool was added and the reaction was incubated for 100 min at 37°C, 20 min at 80°C, and incubated at 4°C until probe set purification. 8 μL of the reaction was analyzed on an 1.25% agarose gel (run time of 8 min at 150 volts) to confirm the probe set length. Probe sets of 300–700 nt were used for the study. The 37°C extension time was increased or decreased (from 100 min) based on desired concatemer length. The probe set was purified using MinElute PCR Purification Kit (Qiagen, cat. #28004) following manufacturer's protocol. The probe set was eluted in 25 μL UltraPure Water and the concentration was analyzed by NanoDrop on the ssDNA setting (Thermo Fisher Scientific). Probe sets with ssDNA concentration ranging from 200 to 500 ng/μL, depending on the hairpin, were used for this study.

Fluorescent oligos were ordered from IDT with 5' modification of either AlexaFluor 488, ATTO 550, ATTO 590, or ATTO 633. The sequences for fluorescent oligos and hairpins are included in Supplementary Files. Working dilutions of the hairpins (5 μM) and oligos (10 μM) were made by diluting in IDTE pH 7.5 and stored at −20°C. Working dilutions of the fluorescent oligos (10 μM) were made by diluting in UltraPure Water and stored at −20°C.

## Fluorescent oligo stripping for multiplexed Probe-Seq

Cells were incubated in 50% formamide solution (diluted in PBS) for 5 min at RT. The cells were then centrifuged at 2000 xg for 5 min at RT. The cells were resuspended in 1 mL of PBS and centrifuged again at 2000 xg for 5 min at RT. Hybridization of new fluorescent oligos was carried out as described above.

## Stain Index calculation

The Stain Index (SI) was calculated by measuring the geometric mean of the positive and negative populations as well as the standard deviation of the negative population using FlowJo. The SI was calculated as follows: $(Geo.\ Mean_{POS} - Geo.\ Mean_{NEG}) / (2 \times SD_{NEG})$.

## Live cell RNA sequencing

In vivo retina electroporation was carried out at P2 as described previously (*Matsuda and Cepko, 2007*). Two plasmids, bpCAG-mTagBFP and Grik1$^{CRM4}$-GFP (*Kishi et al., 2019*), were electroporated simultaneously at a concentration of 1 µg/µL per plasmid. Retinas were harvested at P40. The electroporated and unelectroporated regions were processed separately. The electroporated region was dissociated as described above, and BFP$^+$/GFP$^+$ cells were FACS isolated into Trizol (Thermo Fisher Scientific, cat. #15596026). Cells from the unelectroporated region were used for Probe-Seq as described above. The RNA from the cells in Trizol was extracted following the manufacturer's protocol. The RNA-sequencing libraries were generated using the SMART-Seq v.4 and Nextera XT kits as described above.

## Histology and SABER-FISH

For the mouse retina, adult CD1 mouse retinas were dissected and fixed in 4% PFA for 20 min at RT. The fixed retinas were cryoprotected in 30% sucrose (in PBS). Once submerged, the samples were embedded in 50%/15% OCT/Sucrose mixture in an ethanol/dry ice bath and stored at −80°C. The retinas were cryosectioned at 15 µm thickness. For the human eye, formalin-fixed human postmortem eyes were obtained from Restore Life USA. Patient DRLU101818C was a 54-year-old male with no clinical eye diagnosis and the postmortem interval was 4 hr. Patient DRLU110118A was a 59-year-old female with no clinical eye diagnosis and the postmortem interval was 4 hr. A square (~1 cm x ~ 1 cm) of the human retina was cryoprotected, embedded, and cryosectioned as described above. For the *Drosophila* gut, the midgut was fixed in 4% PFA for 30 min at RT. For the chick retina, the central region of developing HH28 chick retina that contained the developing high acuity area was excised and fixed in 4% PFA for 20 min at RT. The retina was then cryoprotected, embedded as described above, and cryosectioned at 50 µm thickness.

SABER-FISH of retinal sections was carried out on Superfrost Plus slides (Thermo Fisher Scientific, cat. #1255015) using an adhesive hybridization chamber (Grace Bio-Labs, cat. #621502). For *Drosophila* guts, wholemount SABER-FISH was performed in microcentrifuge tubes. Retinal sections were rehydrated in PBS for 5–10 min to remove the OCT on the slides. Subsequently, sections were completely dried to adhere the sections to the slides. Once dry, the adhesive chamber was placed to encase the sections. For both retinal sections and wholemount *Drosophila* guts, the samples were incubated in 0.1% PBS/Tween-20 (MilliporeSigma, cat. #P9416) for at least 10 min. The PBST was removed, and the samples were incubated with pre-warmed (43°C) 40% wHyb (2x SSC, 40% deionized formamide, 1% Tween-20, diluted in UltraPure Water) for at least 15 min at 43°C. The 40% wHyb was removed, and the samples were then incubated with 100 µL of pre-warmed (43°C) probe mix (1 µg of probe per gene, 96 µL of Hyb1 solution (2.5x SSC, 50% deionized formamide, 12.5% Dextran Sulfate, 1.25% Tween-20), diluted up to 120 µL with UltraPure Water) and incubated 16–48 hr at 43°C. The samples were washed twice with 40% wHyb (30 min/wash, 43°C), twice with 2x SSC (15 min/wash, 43°C), and twice with 0.1% PBST (5 min/wash, 37°C). The samples were then incubated with 100 µL of the fluorescent oligo mix (100 µL of PBST, 2 µL of each 10 µM fluorescent oligo) for 15 min at 37°C. The samples were washed three times with PBST at 37°C for 5 min each and counterstained with DAPI (Thermo Fisher Scientific, cat. #D1306; 1:50,000 of 5 mg/mL stock solution in PBS) or WGA-405s (Biotium, cat. #290271; 1:100 of 1 mg/mL stock solution in PBS). Cell segmentation and cell calling algorithms were performed as described previously (*Kishi et al., 2019*).

## Imaging

Fluorescent images were acquired with W1 Yokogawa Spinning disk confocal microscope with a 50 µm pinhole disk and 488 nm, 561 nm, and 640 nm laser lines. The objectives used were either Plan Fluor 40x/1.3 or Plan Apo 60x/1.4 oil objectives, and the camera used was Andor Zyla 4.2 Plus sCMOS monochrome camera. Nikon Elements Acquisition Software (AR 5.02) was used for image acquisition and Fiji 2.0.0 (*Rueden et al., 2017*) or Adobe Photoshop CS6 was used for image analysis. SABER-FISH images were acquired as a z-stack and converted to a 2D image by maximum projection in Fiji.

## Data availability

Raw sequencing data and matrices of read counts for the mouse, chick, and *Drosophila* Probe-Seq are available at GEO: GSE135572.

## Code availability

All R scripts used for differential expression analysis are available in Supplementary Files.

## Acknowledgements

We would like to thank former and current members of the Cepko and Tabin Labs for the insightful discussion and feedback. We thank PM Llopis, R Stephansky, and the MicRoN core at Harvard Medical School for their assistance with microscopy. We thank C Araneo, F Lopez, and the Flow Cytometry Core Facility for their assistance with flow cytometry. We thank S da Silva. for providing the image of in situ hybridization for *FGF8* in the developing chick retina. We thank C Cowan and B Roska for providing the list of human retinal cell type-specific markers. We thank K Okonechnikov for assistance with 3' bias measurement. This work was supported by the Howard Hughes Medical Institute (JC, CLC and NP), Edward R and Anne G Lefler Postdoctoral Fellowship (RA), HSCI Internship Program (MDG), and NIH NEI K99/R00 Grant 5K99EY028215 (SWL).

## Additional information

### Funding

| Funder | Grant reference number | Author |
| --- | --- | --- |
| Howard Hughes Medical Institute | | Jiho Choi<br>Norbert Perrimon<br>Constance L Cepko |
| Harvard Medical School | Edward R and Anne G Lefler Postdoctoral Fellowship | Ryoji Amamoto |
| National Eye Institute | 5K99EY028215 | Sylvain W Lapan |
| Harvard Stem Cell Institute | HSCI Internship Program | Mauricio D Garcia |

The funders did not have a role in the direction of the study.

### Author contributions

Ryoji Amamoto, Conceptualization, Methodology, Software, Formal Analysis, Investigation, Writing - original draft, Writing - review and editing; Mauricio D Garcia, Investigation, Writing - review and editing; Emma R West, Conceptualization, Methodology, Formal Analysis, Writing - review and editing; Jiho Choi, Elizabeth A Lane, Norbert Perrimon, Resources, Writing - review and editing; Sylvain W Lapan, Conceptualization, Methodology, Writing - review and editing; Constance L Cepko, Conceptualization, Supervision, Funding acquisition, Project administration, Writing - original draft, Writing - review and editing

### Author ORCIDs

Ryoji Amamoto https://orcid.org/0000-0002-9335-112X
Norbert Perrimon https://orcid.org/0000-0001-7542-472X
Constance L Cepko https://orcid.org/0000-0002-9945-6387

### Ethics

Human subjects: This IRB protocol (IRB17-1781) was determined to be not-human subject research by the Harvard University-Area Committee on the Use of Human Subjects.
Animal experimentation: All animals were handled according to protocols approved by the Institutional Animal Care and Use Committee (IACUC) of Harvard University (Protocol #1695).

Decision letter and Author response
Decision letter https://doi.org/10.7554/eLife.51452.sa1
Author response https://doi.org/10.7554/eLife.51452.sa2

## Additional files

### Supplementary files

• Source code 1. R code for differential expression alaysis.

• Supplementary file 1. List of oligos used to generate gene-specific probe sets, fluorescent oligos, and hairpins.

• Transparent reporting form

### Data availability

Raw sequencing data and matrices of read counts for the mouse, chick, and Drosophila Probe-Seq are available at GEO under accession number GSE135572.

The following dataset was generated:

| Author(s) | Year | Dataset title | Dataset URL | Database and Identifier |
|---|---|---|---|---|
| Ryoji Amamoto, Constance L Cepko | 2019 | Probe-Seq enables transcriptional profiling of specific cell types from heterogeneous tissue by RNA-based isolation | https://www.ncbi.nlm.nih.gov/geo/query/acc.cgi?acc=GSE135572 | NCBI Gene Expression Omnibus, GSE135572 |

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
