## [Decision Letter]

**Acceptance summary:**

This manuscript describes an innovative non-transgenic method called Probe-seq, to purify and transcriptionally profile genetically-defined populations of cells. The protocol involves FACS-sorting cells based upon strong fluorescence signals produced by a new method of fluorescent RNA in situ hybridization. Cells or nuclei collected with this protocol can produce high-quality RNA sequencing data. The method therefore allows deep sequencing of rare cell populations and it compares well to the efficiency of single cell RNA sequencing using the SMART-seq method. The authors show that Probe-Seq can be applied to a variety of tissues from different organisms and to frozen nuclei isolated from archival material. The method described in this paper can therefore have many applications and will be of interest to large numbers of biologists from diverse fields.

**Decision letter after peer review:**

Thank you for submitting your manuscript "Probe-Seq enables transcriptional profiling of specific cell types from heterogeneous tissue by RNA-based isolation" for consideration by *eLife*. Your article has been reviewed by three peer reviewers, and the evaluation has been overseen by Marianne Bronner as Senior Editor and Francois Guillemot as Reviewing Editor. Sydney Shaffer (Reviewer #3) has agreed to reveal his identity. The reviewers have discussed with one another and the Reviewing Editor has drafted this decision to help you prepare a revised submission.

Summary:

This manuscript provides an innovative, non-transgenic method to purify genetically defined populations of cells from a variety of tissues in diverse organisms. The reviewers are positive about the manuscript but raise several issues that should be addressed in a revision, including:

1) What is the purity of populations FAC-sorted based on their SABER signal?

2) How much does the SABER procedure affect the quality of RNA?

3) The data presented suggest that 12 tiling oligonucleotides are not sufficient for a confident separation of gene-positive and negative populations for highly expressed genes.

The original reviews are included below.

Reviewer #1:

In this manuscript, Amamoto et al. describe an innovative non-transgenic method, Probe-seq, to purify genetically defined populations of cells. They developed a protocol in which FACS sorting of cells is performed based upon strong fluorescence signals produced by SABER-FISH, a method that allows specific labeling of RNAs. They subsequently demonstrate that cells/nuclei collected under this protocol can produce high-quality RNA sequencing data.

Accordingly, the manuscript shows that Probe-Seq allows deep transcriptome profiling of a rare population of cells and compares well to the efficiency of scRNA using the SMART-seq. Several strengths of this manuscript are notable. First, the isolation of cells via RNA FISH has not been very straightforward, and I find this study a major technical achievement. Second, the authors demonstrate that Probe-Seq can be applied not only to mouse retina, but also to a variety of tissues from different organisms, including human, fly, and chick embryos, as well as nuclear samples. Therefore, this study is likely relevant to a broad biological community. The Materials and methods section of this manuscript is clearly written, and supplementary data provide the necessary information, including DNA sequences, to reproduce their data. Overall, I found the data is well presented and convincing.

One minor weakness of the study is that it's not clear how this method fares with more traditional scRNA-seq given only a few comparisons. Also, while the isolation of even finer cell types using sequential cell sorting is impressive, sequencing data from the sorted population is not presented in this manuscript to support whether this workflow is indeed feasible for RNA-seq. The manuscript will be significantly improved and more impactful if these data can be added.

Reviewer #2:

This methodological paper combines fluorescent in situ hybridization (FISH) and RNA-sequencing for cell sorting and transcriptional analysis of enriched cell populations. The principal advantage of this approach is its versatility, as it can be applied to a variety of tissues and species without the need for transgenesis.

Authors show that the FISH procedure is compatible with the identification of differentially expressed genes in various contexts. Although the approach is interesting, I believe that it would be considerably strengthened by performing a more direct quantification of the consequences of the FISH procedure on RNA quality and subsequent transcriptional analysis. The comparison of the Probe-Seq data with those obtained from freshly dissociated cells following retina electroporation, performed in the first part of the manuscript, is not sufficient. Indeed, cells labelled by the two procedures are not directly comparable, thereby preventing a direct comparison of the transcriptional results. A more direct comparison (i.e. RNA quality/transcriptional differences observed following FAC-sorting, vs. Probe-Seq vs. repeated Probe-Seq) should be performed to validate the approach and demonstrate its full potential.

Please find additional comments below:

- Figure 1 summarizes the procedure workflow as well as main results obtained for the adult mouse retina. It is accompanied by four supplementary figures, some of which containing important information (e.g. Figure 1—figure supplement 3). I would suggest splitting this figure in order to incorporate some of these data within the main body of the manuscript.

- Figures 2 and 3 follow the exact same workflow to demonstrate that the approach can be transposed to frozen post-mortem cells and/or other species. These two figures could be merged, as they essentially convey a similar message.

- Figure 1H and Figure 1—figure supplement 5: although authors conclude that 12 tiling oligonucleotides is sufficient for a confident separation of gene-positive and negative populations, a close look at flow cytometry histograms presented on Figure 1—figure supplement 5 rather suggests that it is insufficient for highly expressed genes (*Grik1*). Please revise.

- Demonstration of ON BC cell enrichment by *GRM6* Probe-Seq is not convincing if one refers to Figure 2E, as markers of ON BC are barely enriched when compared to those of OFF BC cells on the presented heat map.

- Validation of *CDH12* as a marker enriched in *GRM6*-sorted cells is not supported by the in situ hybridization presented in Figure 2F. Indeed, expression of CDH12 can be observed in other cell layers/cell types.

Reviewer #3:

This paper describes a new method for isolating specific cell populations based on gene expression for RNA sequencing. This method uses in situ labeling of specific RNA species using an RNA FISH amplification method called SABER-FISH. The SABER-FISH signal is specific to the gene of interest and sufficiently bright for isolated cells using fluorescently activated cell sorting. They use this strategy to sort their desired cell populations that can then be analyzed by bulk RNA seq to give a more in-depth expression profile of specific types of cells. To show the broad applicability of this method, they apply it on mouse, human, *Drosophila*, and chick tissue, and they also show that it is compatible with nuclei from frozen tissue as well. Overall, I believe this is a very practical and useful method for getting in depth and high-quality RNA-seq data on a specific population of cells within a tissue. I can imagine many applications and biological questions where this method will be really useful!

1) In general, it would be helpful to know the expected purities from these SABER + FACS protocols. The authors show images after sorting the populations of cells (particularly in Figure 1E), but it would be useful to show more quantification of these sorted cells. What fraction of the sorted cells that are supposed to show a particular marker combination actually have that combination of SABER signal when imaging (the *Vsx2*^+^/*Grik1*^-^ and *Vsx2*^+^/*Grik1*^+^ samples)? This helps give an idea of how pure these populations are prior to bulk-seq. Do these fractions match what is expected from previous literature/data?

2) It would be helpful to provide some quantification for the data shown in Figure 2F. For example, how many of the *GRM6^+^* cells are also *CDH12^+^* and vice versa? In these images, why does *GRM6* have larger (and more irregularly sized) spots than *CDH12*?

3) Do the authors know why *Neto1* (lowest expression) has the best stain index and *Grik1* (highest expression) has the worst when using 12 oligos? Is this something systematic about the method or just a particular finding with these genes? How was the cutoff of SI = 2 selected for this analysis? For people who would want to use this method, is SI > 2 what is being recommended?

4) For the majority of FACS plots, the fluorescence of the gene of interest is plotted against autofluorescence. Is this a proxy for cell size or side scatter? Are there any specific advantages to using autofluorescence the readers should know about?

---

## [Author Response]

Summary:This manuscript provides an innovative, non-transgenic method to purify genetically defined populations of cells from a variety of tissues in diverse organisms. The reviewers are positive about the manuscript but raise several issues that should be addressed in a revision, including:1) What is the purity of populations FAC-sorted based on their SABER signal?

To determine the purity of populations FAC-sorted based on their SABER signal, we placed them on a microscope slide after FACS isolation. We then quantified the percentage of cells that had *Vsx2* puncta. This analysis showed 92.3 ± 0.6% purity based on three individual sorts. In the text, we have added, “To determine the purity of populations isolated using FACS, based on their SABER signal, the cells were placed on a microscope slide after FACS, and the percentage of cells that had *Vsx2* puncta was quantified. This analysis showed 92.3 ± 0.6% had *Vsx2* puncta, based on three individual sorts.”

2) How much does the SABER procedure affect the quality of RNA?

To answer this question, we measured the 3’ bias using the Qualimap software. Measurement of 3’ bias is a more reliable method to assess RNA quality, relative to the RNA Integrity Number (RIN). The RIN method measures RNA quality based only on ribosomal RNA, while the 3’ bias method takes into account the 10,000 most highly expressed genes. We used 100,000 cells from the *Grik1^-^*Live cell population (FACS isolated and RNA extracted by Trizol) and 100,000 cells from the *Grik1^-^*Probe-Seq population. The cell composition and number were nearly identical in these two populations. As expected, the gene body coverage of the Live cell population showed no 3’ bias, suggesting little to no RNA degradation. In comparison, the Probe-Seq gene body coverage showed 3’ bias, likely corresponding to a RIN score of 4-6 (Figure 1—figure supplement 4). In the Results section, we added, “To measure the RNA quality of the Live cells and Probe-Seq cells, 100,000 GFP^-^ Live cells (n=3) and 100,000 *Grik1^-^*Probe-Seq cells (n=3) were collected. Based on the gene body coverage of the 10,000 most highly expressed genes, a slightly higher 3’-5’ bias was observed for the RNA originating in the Probe-Seq population (1.02 ± 0.02) compared to the Live cell population (0.90 ± 0.01), indicating mild degradation of RNA with the Probe-Seq protocol. Based on the gene body coverage graph, the level of degradation for the Probe-Seq population would project to a RIN score of approximately 4-6(Sigurgeirsson, Emanuelsson and Lundeberg, 2014) (Figure 1—figure supplement 4).”

3) The data presented suggest that 12 tiling oligonucleotides are not sufficient for a confident separation of gene-positive and negative populations for highly expressed genes.

We agree that the gene-positive events using 12 tiling oligonucleotides were difficult to separate from the gene-negative events when using a histogram. Therefore, we added a new figure panel (Figure 1—figure supplement 5D) showing the 2-dimensional flow cytometry plots which more clearly show the separation between negative and positive populations. In the text, we added, “However, with an SI cutoff of 2, 12 oligos were sufficient for confidence in the separation of gene-positive and negative populations. This was evident only when the events were displayed in a 2-dimensional flow cytometry plot (Figure 1—figure supplement 5).”

The original reviews are included below.

In addition to the necessary revisions above, we chose to address the following concerns from the individual reviews as we believe that these changes further improve the manuscript.

Reviewer #2:[…] - Validation of CDH12 as a marker enriched in GRM6-sorted cells is not supported by the in situ hybridization presented in Figure 2F. Indeed, expression of CDH12 can be observed in other cell layers/cell types.

The strong fluorescence was detected in the Outer Plexiform Layer, which we believe is due to autofluorescence, as the signals were not punctate. We have added an asterisk in the figure to indicate this.

Reviewer #3:[…] 3) Do the authors know why Neto1 (lowest expression) has the best stain index and Grik1 (highest expression) has the worst when using 12 oligos? Is this something systematic about the method or just a particular finding with these genes? How was the cutoff of SI = 2 selected for this analysis? For people who would want to use this method, is SI > 2 what is being recommended?

It is unclear to us why *Neto1,* which had the lowest expression of the three genes tested, gave the highest staining index with 12 oligos. We suspect that this is a particular finding with these genes, suggesting that the staining index is not always linearly correlated with the number of tiling oligonucleotides. It is also possible that if we picked 12 different oligonucleotides (out of 48), then the staining index could differ. SI = 2 cutoff was determined empirically as we were able to see a distinct population with SI = 2.4. However, with such a low staining index, we were only able to see a clear separation between negative and positive populations using a 2D plot instead of a histogram (Figure 1—figure supplement 5).

4) For the majority of FACS plots, the fluorescence of the gene of interest is plotted against autofluorescence. Is this a proxy for cell size or side scatter? Are there any specific advantages to using autofluorescence the readers should know about?

We added the following to Materials and methods, “The empty channel was used to determine autofluorescence, and the events that displayed high intensity for both the channel of interest and the empty channel were deemed negative as they may be events with high autofluorescence in all channels.”

Additional Changes:

We have also re-analyzed the *Drosophila* Probe-Seq dataset using an updated single cell RNA sequencing dataset (Hung et al., 2019) that now includes significantly more cells. However, the conclusion remains the same as the previous analysis.